# Ranking versus rating in peer review of research grant applications

**Robyn Tamblyn**[ID][1,2,3]*, **Nadyne Girard**[1], **James Hanley**[2], **Bettina Habib**[1], **Adrian Mota**[4], **Karim M. Khan**[4,5,6], **Clare L. Ardern**[ID][7,8]

**1** Clinical and Health Informatics Research Group, McGill University, Montreal, Canada, **2** Department of Epidemiology, Biostatistics and Occupational Health, McGill University, Montreal, Canada, **3** Department of Medicine, McGill University Health Center, Montreal, Canada, **4** Canadian Institutes of Health Research (CIHR), Ottawa, Canada, **5** Department of Family Practice, University of British Columbia, Vancouver, Canada, **6** School of Kinesiology, University of British Columbia, Vancouver, Canada, **7** Department of Physical Therapy, University of British Columbia, Vancouver, Canada, **8** Sport and Exercise Medicine Research Centre, La Trobe University, Melbourne, Australia

* robyn.tamblyn@mcgill.ca

**Data Availability Statement:** The datasets analyzed in this study are held by the Canadian Institutes for Health Research (CIHR) and are not publicly available due to privacy and legal restrictions. Researchers wishing to obtain access

## Abstract

The allocation of public funds for research has been predominantly based on peer review where reviewers are asked to rate an application on some form of ordinal scale from poor to excellent. Poor reliability and bias of peer review rating has led funding agencies to experiment with different approaches to assess applications. In this study, we compared the reliability and potential sources of bias associated with application rating with those of application ranking in 3,156 applications to the Canadian Institutes of Health Research. Ranking was more reliable than rating and less susceptible to the characteristics of the review panel, such as level of expertise and experience, for both reliability and potential sources of bias. However, both rating and ranking penalized early career investigators and favoured older applicants. Sex bias was only evident for rating and only when the applicant's H-index was at the lower end of the H-index distribution. We conclude that when compared to rating, ranking provides a more reliable assessment of the quality of research applications, is not as influenced by reviewer expertise or experience, and is associated with fewer sources of bias. Research funding agencies should consider adopting ranking methods to improve the quality of funding decisions in health research.

## Introduction

For decades, the allocation of public funds for research has been predominantly based on peer review. In complex areas of endeavor, such as the medical sciences, it is assumed that the assessment of quality and potential impact is best done by peers who have the in-depth knowledge needed to evaluate the quality of both the scientific team and the proposed research. Although peer review is supported by the academic community [1], it has many vocal critics [2, 3]. There is evidence that peer review is conservative and less likely to fund higher risk, innovative projects, or those that involve a multidisciplinary team of investigators [4–8].

to these data need to contact the Vice-President of Research Programs-Operations at CIHR (christian. baron@cihr-irsc.gc.ca) to obtain approval to access de-identified data on foundation funding program applications submitted between 2014 and 2017.

**Funding:** Funding was provided by the Canadian Institutes of Health Research (CIHR). The study sponsor approved the use of the data, and the original manuscript as per CIHR policy. CIHR had no role in the design of the study, the analysis of interpretation of the data, or the the review and approval of the final published manuscript.

**Competing interests:** The authors have declared that no competing interests exist.

Perhaps the most distressing aspect of peer review for applicants is the low level of agreement among experts about the quality of the application. The reliability of peer review rating varies from 0.20 to 0.61 (intra-class correlation coefficient) in different studies [9–14], being somewhat higher for the basic medical sciences (0.41) than the applied sciences (0.32) [11]. The variability in rating translates into a lack of consistency in decision-making about which applications should be funded. Depending on the subset of reviewers selected to review an application, variability in scoring alters the funding decision in approximately 17% to 35% of proposals [10, 15–19].

Most funding agencies ask reviewers to rate an application on some form of ordinal scale from poor to excellent, for individual criteria, such as research approach, as well as an overall application score. This form of rating, referred to as "absolute judgment", requires a reviewer to judge an application against an internalized concept of an "ideal" application. Experienced reviewers who have reviewed hundreds of applications would be expected to have a more robust and stable concept of the "ideal" application than less experienced reviewers [20]. This hypothesis was indirectly supported by a small study of public health reviewers where training improved reliability from an intra-class correlation of 0.61 to 0.89, with the effect being greatest for inexperienced reviewers [12]. In contrast to "absolute" judgment, "relative" judgements do not rely on an internal concept of the ideal application [20]. Instead, reviewers are asked to rank applications from the highest quality to the lowest. Relative judgments do not rely on a stable internalized concept of the ideal application. They are expected to be more reliable, as they are less sensitive to differences in the experience of the reviewers, as well as systematic differences in the harshness or leniency of the reviewer [9, 18, 21] and thus should reduce random variation in judgements. Even within-person variation in judgement appears to be more stable with ranking. For example, comparative ranking of preferences is 20% more stable over time than rating of preferences [22].

Ranking compared to rating also appears to reduce bias in judgements in cross-national studies [23]. This finding is of particular interest as there is evidence of systematic bias in peer review of grant applications. Female applicants who are as qualified and productive as male applicants, are scored systematically lower in both postdoctoral and grant applications [11, 24–26], and receive less NIH funding [27]. Similar systematic biases in judgement are observed for black compared to white applicants in NIH competitions, biases that appear to be related to differences in how individual criteria are rated [28, 29].

An opportunity to compare the reliability and potential for bias of rating and ranking occurred when the Canadian Institutes of Health Research (CIHR) introduced a new funding program and scoring approach to support the research of leading scientists and rising stars: the foundation funding program [30]. In this program, reviewers scored and then ranked the applicants in two phases: first the quality of the applicant, and then for the highest-ranking applicants, the quality of the research approach. We evaluated whether ranking as compared to rating improved the reliability of peer review and reduced potential sources of bias for two distinct aspects of the quality of the application.

## Methods

### Design, population and data sources

A historical cohort study was conducted that included all applications submitted to the CIHR foundation funding program between 2014 when it was first launched to 2017. For each application, the principal investigator's CIHR number, sex, age, institution, co-investigators, application title, scientific domain (biomedical, clinical, health services, population health), requested budget and duration of funding were retrieved. For each application, we also

identified the CIHR number of the review committee chair, all reviewers that were approached to review, and the reviewers' self-rated conflict of interest, and expertise to review. The study was approved by CIHR senior executive management and CIHR legal counsel. All data were stored on secured servers at McGill University and after linkages, all nominal data were removed.

## The foundation funding program

The objective of the foundation funding program was to fund the research programs of leading scientists, both at the early and later career stages, for a period of 5–7 years to provide sustained support and flexibility for innovative, high impact research. In the three-phase review process, top ranked applications in phase 1 were invited to apply to phase 2. The first two phases were conducted remotely through a web platform. In the third phase, applications and reviews were discussed at a multidisciplinary in-person meeting where a decision was made about funding. There were no fixed panels of reviewers. Instead, at each phase of this application-centric review, the application was assigned a set of expert reviewers from the CIHR College of Reviewers, a pool of over 4,000 Canadian and International Scientists who had previously reviewed CIHR funding applications [31], by scientific chairs. A different set of chairs and reviewers was selected for each phase.

## Review process and assessment of quality

In the first phase, "virtual" chairs were assigned a set of 1 to 50 applications. The chairs selected 4–5 expert reviewers per application from the College of Reviewers. Reviewers were assigned between 5 to 20 applications to review, and reviewers received their assignments from 1 to 11 chairs. Reviewers assessed the caliber of the applicant using four criteria: vision, impact, productivity, and leadership. Each criterion was rated based on the applicant's curriculum vitae and a two-page summary of their contributions in each of these domains. Ratings were done using letter grades (poor, fair, good, excellent, excellent+, excellent++, outstanding, outstanding+, outstanding++). Letter grades were converted to a score from 0 (poor) to 28 (outstanding++) per criterion with a maximum score of 112. All the applications rated by a given reviewer were then ranked by score. The reviewer then adjusted the application rank to reflect their judgement of the best to the worst applications and to eliminate ties when applicable. As reviewers rated a different number of applications, ranks were standardized by converting them to percentiles. The mean of all reviewers' percentile ranks was the final score for an application.

In the second phase, a second set of "virtual" chairs were assigned a set of 1 to 45 phase 2 applications, and they selected 4–5 expert reviewers from the College of Reviewers. Reviewers assessed the quality of the program of research according to the following criteria: research concept, research approach, expertise, mentorship, and environmental support based on a 10-page application. Each criterion was graded using the same letter grades as in phase 1, with a maximum score of 140. However, the criteria were weighted differently, with 50% of the weight assigned to the research concept and approach, 40% to expertise and mentoring, and 10% to environmental support. Similar to phase 1, letter grades were converted to a score that was used to establish a preliminary rank order. The reviewer then modified the ranked applications from best to worst. The reviewer-adjusted ranks were converted to percentiles, and the mean of all reviewer ranks was the final score for the application. The 5 reviewers for an application did not necessarily have the same applications to review so it was assumed that they would, on average, have an equivalent distribution of poor, good and excellent applications.

In both phase 1 and 2, applications were flagged for asynchronous discussion if there was differences in scores and ranks of greater than one standard deviation, or if the virtual chair identified comments/ questions raised by one or more reviewers that warranted discussion.

## Outcomes

To assess the reliability and potential bias in rating and ranking methods of assessment, we used the overall rating and percentile rank scores assigned to an application by the different sets of reviewers and review processes in phase 1 and 2. In addition, for the rating method, we assessed reliability using reviewers' assessment of the individual criteria to determine if there was greater agreement among reviewers for some criteria than others. To assess potential predictors of reliability, we calculated the between-rater variance for an application and used the log of this variance as the outcome variable. As we had no "gold standard" assessment of the quality of the application, we used measures of the applicant's scientific productivity to assess potential predictors of bias among applicants with equivalent productivity.

## Potential predictors of reliability and bias

**Applicant characteristics.** Applicant *age*, *sex*, and *institution* were retrieved from the demographic information provided in the application. Applicant *scientific productivity* was measured using the *H-index*. The *H-Index* measures the impact of the applicant's cumulative research contributions based on citations, allowing unbiased comparison among applicants competing for the same resources [32]. To calculate the *H-index*, we used the principal applicant's first, middle, and last name and institution to retrieve all publications from the Web of Science database where the applicant was listed as an author up to and including the year in which the application was submitted. For each publication, we retrieved the citation reports by linking the ISSN of the journal to the Journal Citation Record file. When there was no recorded ISSN, we used the full and abbreviated journal name to make the link. The applicant was classified as an *early career investigator* if the application was submitted within 5 years of their first university/institute research appointment, as a dedicated stream of funding was available for this group. As the H-index can be positively biased by the number of collaborators on an applicant's publications [33], we also counted the number of unique collaborators associated with the applicants publications.

**Application characteristics.** The *year* in which the application was submitted was included as the foundation scheme was a new program and the review process was expected to improve with applicant and reviewer experience. As there was a change in policy after the first competition in 2014 to provide equivalent opportunity for male and female applicants to progress to phase 2, we classified the application submission year as 2014 or after 2014 [25]. The *content domain of the application* was included, as reliability tends to be better for biomedical applications compared to clinical, health services and population health applications [11, 13].

**Review characteristics.** As the role of the chair is to facilitate unbiased high quality reviews, discussion and consensus on the quality of an application, where possible, we measured, for each application, three characteristics of the chair that may influence their performance. These included: 1) *the number of reviewers the chair was responsible for*, 2) *the number applications the chair was assigned*, and 3) *the number of years of experience the chair had since 2000 as a reviewer*. With respect to reviewers, for each application, we measured: 1) *the number of reviewers assigned to an application*, 2) the *mean number of years of review experience* since 2000 of all reviewers assigned to an application, as greater experience should be associated with better reliability for rating [20], 3) *the sex mix of reviewers*, as female reviewers score systematically lower than male reviewers so a mix of male and female reviewers may decrease

reliability [11], 4) *the proportion of reviewers with high self-assessed expertise to review* an application, 4) *the proportion of reviewers whose prior applications were in the same scientific domain* as the applicant, and 5) the *workload of the reviewers* of an application, measured as the mean of the number of applications each reviewer was assigned to review. With respect to the review process, we measured whether there was *on-line discussion (yes/no)* using data retrieved from the on-line review system. We also measured the proportion of reviewers who declined to review an application because of conflicts of interest, as conflicts in the review panel have been associated with higher scores [5, 11, 24, 34, 35], a phenomenon that may not apply when reviews are done virtually rather than in-person.

### Data analysis

To assess reliability of rating and ranking, we estimated the intra-class correlation coefficient (ICC) for the overall rating and percentile ranking scores separately for applications in the phase 1 and phase 2 review stages. The ICC for ratings of individual criteria were also estimated. Bootstrapping was used to estimate 95% confidence intervals. To estimate the association between applicant, application and review characteristics and within-rater variance of application scores, we used generalized estimating equation multiple linear regression to account for clustering of applications within applicants (i.e. repeated submissions). Application was the unit of analysis. To assess potential bias in review by rating compared to ranking, we created two multivariate models, one using overall mean rating of the application as the outcome and the second using overall mean percentile ranking. The H index of the principal applicant and the log of the number of collaboraters were included in the model to assess the impact of other attributes of the applicant, application and review process that may have influenced scores among applicants with equivalent scientific productivity. As the H-index in prior research has been shown to have a non-linear relationship to the application score [11], we tested whether including the quadratic term for the H-index improved model fit. All measured characteristics of the applicant, application and review process were included in the model. Among scientists with equivalent scientific productivity, we tested the hypothesis that the overall rating and ranking score may be modified by applicant sex or age by including the two-way interaction terms (H-index* applicant sex, H-index*applicant age), in separate multivariate models. All analyses were conducted using SAS, version 9.41M5.

## Results

### Study population characteristics

In the 4 years the foundation program was offered, 3,156 applications were submitted by 2,249 investigators. Female investigators accounted for 32.6% of applications and 38.3% of applicants were aged 41 to 50 years (Table 1). Early career investigators submitted 29.8% of applications. The mean H-index of applicants was 13.5 (SD 9.9), and 24.3% had an H-index of greater than 19 in the year they applied. The mean number of collaboraters in phase 1 was 507.5 (SD 663.7) and in phase 2 it was 763.1 (SD 860.4). Most applications were submitted in 2014 (42.6%) or 2015 (28.8%) and were in the basic science domain (58.5%).

Virtual Chairs were responsible for a mean of 23.3 applications and 31.1 reviewers in phase 1, and 13.6 applications and 28.5 reviewers in phase 2 (Table 1). The majority of applications in phase 1 (66.9%) and phase 2 (75.0%) were reviewed by 5 reviewers. Each reviewer had a mean of 12.6 (range: 5–20) applications to review in phase 1, and 9.5 (range: 4–14) in phase 2. The majority of reviewers had 2 to 3 years of CIHR review experience. In phase 1, 73.0% of applications had reviewers with less than 60% higher expertise to review; 55.4% in phase 2. There was on-line discussion for 19.2% of applications in phase 1 and 36.1% of applications in phase 2.

**Table 1. Applicant, application, and review characteristics.**

| | Phase 1 Leadership & Vision | Phase 2 Research Methods | | Phase 1 Leadership & Vision | Phase 2 Research Methods |
|---|---|---|---|---|---|
| | N (%) | N (%) | | N (%) | N (%) |
| **Applicant Characteristics** | | | **Reviewer Workload** | | |
| **Sex** | | | Chair–# of reviewers, mean (SD) | 31.1 (13.8) | 28.5 (17.1) |
| Male | 2126 (67.4%) | 794 (72.4%) | Chair–# of applications, mean (SD) | 23.3 (10.9) | 13.6 (10.2) |
| Female | 1030 (32.6%) | 302 (27.6%) | Reviewer–# of applications, mean (SD) | 12.6 (3.2) | 9.5 (2.1) |
| **Age** | | | **Reviewer characteristics** | | |
| <40 years old | 807 (25.6%) | 201 (18.3%) | **CIHR Experience** | | |
| 41–50 years old | 1208 (38.3%) | 380 (34.7%) | < = 2 years | 442 (14.0%) | 315 (28.7%) |
| 51–60 years old | 830 (26.3%) | 347 (31.7%) | >2–3 | 1724 (54.6%) | 485 (44.3%) |
| >60 years | 311 (9.9%) | 168 (15.3%) | >3–4 | 810 (25.7%) | 214 (19.5%) |
| **Early Career Investigator** | | | >4 | 180 (5.7%) | 82 (7.5%) |
| Yes | 939 (29.8%) | 219 (20.0%) | **% of Female Reviewers** | | |
| No | 2217 (70.2%) | 877 (80.0%) | <50% | 2010 (63.7%) | 814 (74.3%) |
| **H-Index** | | | 50%-79% | 875 (27.7%) | 225 (20.5%) |
| < = 6 | 792 (25.1%) | 137 (12.5%) | 80% more | 271 (8.6%) | 57 (5.2%) |
| >6-< = 12 | 854 (27.1%) | 213 (19.4%) | **Mean age of the reviewer (SD)** | 48 (9.5) | 50 (9.7) |
| >12-< = 19 | 744 (23.6%) | 280 (25.5%) | **Review Process** | | |
| >19 | 766 (24.3%) | 466 (42.5%) | **% Reviewers with High Expertise** | | |
| **N of Collaborators** | | | >80% | 317 (10.0%) | 281 (25.6%) |
| < = 122 | 794 (25.2%) | 137 (12.5%) | 60%-<80% | 536 (17.0%) | 208 (19.0%) |
| >122-< = 302 | 786 (24.9%) | 195 (17.8%) | <60% | 2303 (73.0%) | 607 (55.4%) |
| >302-< = 643 | 789 (25.0%) | 317 (28.9%) | **% Reviewers with Applications in Same Domain** | | |
| >643 | 787 (24.9%) | 447 (40.8%) | >80% | 504 (15.9%) | 310 (28.3%) |
| **Application Characteristics** | | | 60%-<80% | 589 (18.7%) | 145 (13.2%) |
| **Year Submitted** | | | <60% | 2063 (65.4%) | 641(58.5%) |
| 2014 | 1343 (42.6%) | 445 (40.6%) | **On-Line Discussion** | | |
| 2015 | 910 (28.8%) | 260 (23.7%) | Yes | 607 (19.2%) | 396 (36.1%) |
| 2016 | 600 (19.0%) | 228 (20.8%) | No | 2549 (80.8%) | 700 (63.9%) |
| 2017 | 303 (9.6%) | 163 (14.9%) | **Conflicts of Reviewers Approached** | | |
| **Scientific Domain** | | | None | 720 (22.8%) | 35 (3.2%) |
| Biomedical | 1847 (58.5%) | 639 (58.3%) | At least 1 | 2436 (77.2%) | 1061 (96.8%) |
| Clinical | 590 (18.7%) | 223 (20.3%) | | | |
| HSR | 315 (10.0%) | 101 (9.2%) | | | |
| PPH | 395 (12.5%) | 132 (12.0%) | | | |

CIHR = Canadian Institutes of Health Research, SD = standard deviation, HSR = Health services research, PPH = Population & public health

Phase 1, Leadership & Vision: Number of principal applicants = 2,249, Number of applications = 3,156

Phase 2, Research Methods & Environment: Number of principal applicants = 863, Number of applications = 1,096

## Application rating and ranking characteristics

The mean overall score for phase 1 was 85.1 out of 128 and percentile rank was 49.4%, and for phase 2 it was 115.3 and 50.0%, respectively (Table 2). The correlation between the overall rating and rank in phase 1 was 0.83, and in phase 2 it was 0.78. The highest scoring criterion in

**Table 2. Application scores and percentile ranks.**

| Application Score | Mean (SD) |
|---|---|
| **Phase 1 Leadership & Vision** | |
| **Initial Score** | 85.1 (10.5) |
| Leadership | 21.9 (3.0) |
| Vision | 20.2 (3.8) |
| Productivity | 21.6 (3.1) |
| Impact | 21.4 (3.1) |
| **Percentile Rank** | 49.4 (16.6) |
| **Phase 2 Research Methods** | |
| **Initial Score** | 115.3 (10.8) |
| Research Concept | 22.2 (3.1) |
| Research Approach | 21.0 (3.4) |
| Expertise | 24.2 (2.5) |
| Mentorship | 23.2 (2.9) |
| Environment | 24.7 (2.3) |
| **Percentile Rank** | 50.0 (21.7) |

phase 1 was for leadership (mean 21.9) and in phase 2 it was for the environment (mean 24.7). The item to total correlations for criterion rating in phase 1 was (Chronbach alpha) 0.91, and in phase 2 it was 0.86.

## Reliability of rating and ranking

The overall reliability of assessment (ICC) for phase 1 was 0.54 for rating and 0.59 for ranking (Table 3). For individual criteria rated in phase 1, the highest ICC was for productivity (0.50) and the lowest was for vision (0.35). In phase 1, on average, reviewers had to break ties in overall score for 31.7% (SD 20.0%) of their applications, and 84% of reviewers had to break at least one tie. For Phase 2, the overall reliability was 0.25 for rating and 0.38 for ranking. For the 1096 (34.7%) of applicants invited to apply to phase 2, reliability was highest for research approach (0.23), and lowest for support (0.12). In phase 2 reviewers had to break ties in 28.3% (SD 24.1%) of their applications, and 75.4%of reviewers had to break at least one tie.

## Applicant, application, and review characteristics associated with within rater variance in rating and ranking

In phase 1, the applicant's H-index was significantly associated with rater variance—for each one point increase in the H-index, the rater variance was reduced by -4,19% (p<0.01) in rating and by -2.25% (p<0.01) in ranking (Table 4). Compared to applications in biomedical science, there was significantly greater rater variance of 25.8% (p = 0.01) for ranking but not for rating (8.63%, p = 0.13) for those in applied science. Review characteristics influenced variance of rating but not ranking. Rater variance decreased when a greater proportion of the reviewers had more experience (-9.62%, p<0.01), were in the same scientific domain (-30.05%, p<0.01), and were more likely to have conflicts (-20.51% p<0.01). A greater number of applications assigned to a reviewer and increasing reviewer age were associated with a significant increase in reviewer variance. In contrast, on-line discussion was the only review characteristic associated with increased rater variance in ranking.

Similar to phase 1, in phase 2, rating compared to ranking was more likely to be influenced by review, applicant, and application characteristics (Table 4). Variance in rating was greater

**Table 3. Reliability (ICC) of rating and ranking.**

| Characteristics | ICC (95% CI) | Variance Between Applications | Rater Variance Within Applications |
|---|---|---|---|
| | | **Phase 1 Leadership & Vision** | |
| **Percentile Rank** | 0.59 (0.58; 0.61) | 3181.10 (3076.72; 3289.27) | 406.49 (394.38; 418.69) |
| **Initial Rating** | 0.54 (0.52; 0.55) | 1010.88 (956.37; 1065.68) | 156.22 (150.61; 162.10) |
| **Rating Criteria** | | | |
| Impact | 0.46 (0.44; 0.48) | 69.16 (65.26; 73.01) | 13.63 (13.10; 14.20) |
| Leadership | 0.48 (0.47; 0.50) | 74.66 (70.41; 78.74) | 13.74 (13.15; 14.34) |
| Productivity | 0.50 (0.48; 0.51) | 78.00 (73.50; 82.62) | 13.87 (13.32; 14.45) |
| Vision | 0.35 (0.33; 0.37) | 70.83 (67.07; 74.26) | 19.89 (19.23; 20.54) |
| | | **Phase 2 Research Methods** | |
| **Percentile Rank** | 0.38 (0.35; 0.40) | 2614.48 (2445.57; 2773.62) | 670.83 (641.82; 700.09) |
| **Initial Rating** | 0.25 (0.22; 0.28) | 501.15 (458.34; 544.38) | 191.28 (177.63; 204.82) |
| **Rating Criteria** | | | |
| Expertise | 0.19 (0.16; 0.22) | 22.65 (20.71; 24.67) | 10.62 (9.77; 11.51) |
| Mentorship | 0.17 (0.14; 0.21) | 26.55 (23.65; 29.69) | 13.22 (12.36; 14.17) |
| Research Approach | 0.23 (0.20; 0.26) | 43.78 (40.42; 47.46) | 17.98 (16.86; 19.17) |
| Research Concept | 0.21 (0.18; 0.24) | 37.07 (33.83; 40.22) | 16.09 (15.02; 17.20) |
| Support | 0.12 (0.09; 0.15) | 15.44 (13.73; 17.25) | 9.37 (8.62; 10.17) |

ICC = Intra-class correlation coefficient

for female applicants and those whose applications were in the applied compared to the bio-medical science domain. A higher H-index was associated with a reduction in variance, but only significantly so for rating compared to ranking. Reductions in the variance for both rating and ranking were seen in the subsequent 2015–2017 competitions compared to 2014, although only significant for rating. Variance in rating and ranking was reduced when a greater proportion of reviewers had high expertise. On-line discussion was a marker of disagreement, being associated with increased variance for both rating and ranking.

## Applicant, application and review characteristics associated with systematic bias in rating and ranking

In the assessment of possible bias, we found the expected positive association between the applicant's H-index and overall application rank and score for both phase 1 and 2 (Table 5). The quadratic H-index term was significant, indicating that the H-index was associated with higher application scores, the slope of the increase decreasing at the upper end of the distribution. After adjusting for H-index, number of collaboraters and other application and review characteristics, older applicants received higher scores and ranks in phase 1 but applicant age had no impact in phase 2. Early career investigators received lower scores and ranks in both phase 1 and 2, although not significantly so for ranking in phase 1. Female applicants received significantly lower ratings in phase 1, but there was no impact of applicant sex in phase 2 for either rating or ranking. Of interest, there was a significant interaction between H-index and applicant sex in phase 1 rating (Fig 1). Female applicants were rated lower than male applicants with lower values of the H-index, with rating of female applicants being equivalent or higher than male applicants at the upper end of the H index distribution. None of the other hypothesized interactions between applicant and reviewer sex were significant.

Review characteristics influenced both the overall rating of an application and percentile rank in phase 1 and 2 (Table 5). With respect to workload, the number of reviewers the chair

**Table 4. Association between applicant, application and review characteristics and rater variance for the two methods: Rating and ranking.**

| | Phase 1 Leadership & Vision | | | | Phase 2 Research Methods | | | |
| --- | --- | --- | --- | --- | --- | --- | --- | --- |
| | Variance Rating | | Variance Percentile Rank | | Variance Rating | | Variance Percentile Rank | |
| | Percent change (95% CI) | P-Value | Percent change (95% CI) | P-Value | Percent change (95% CI) | P-Value | Percent change (95% CI) | P-Value |
| **Applicant Characteristics** | | | | | | | | |
| H index | -4.19 (-4.79; -3.58) | <0.01 | -2.45 (-3.95; -0.93) | <0.01 | -1.40 (-2.1; -0.7) | <0.01 | -0.65 (-2.25; 0.98) | 0.43 |
| Total N collaborators (log) | 2.43 (1.67; 3.19) | <0.01 | 1.41 (0.11; 2.74) | 0.03 | 1.13 (-0.1; 2.3) | 0.06 | 0.41 (-1.09; 1.93) | 0.59 |
| PI age | -1.15 (-1.76; -0.54) | <0.01 | -0.65 (-1.69; 0.40) | 0.22 | -0.75 (-1.6; 0.1) | 0.09 | -1.44 (-2.85; -0.00) | 0.05 |
| Sex | | | | | | | | |
| Male | Ref | | Ref | | | | | |
| Female | 4.51 (-4.84; 14.79) | 0.36 | -13.17 (-26.63; 2.77) | 0.10 | 15.95 (0.2; 34.2) | 0.05 | 8.59 (-5.99; 25.43) | 0.26 |
| Early Career | | | | | | | | |
| No | Ref | | Ref | | | | | |
| Yes | 9.30 (-2.19; 22.13) | 0.12 | 0.63 (-17.34; 22.51) | 0.95 | -8.23 (-24.97; 12.25) | 0.4 | -21.28 (-41.0; 4.9) | 0.1 |
| **Application Characteristics** | | | | | | | | |
| Scientific Domain | | | | | | | | |
| Basic Science | Ref | | Ref | | | | | |
| Applied Science | 8.60 (-2.19; 20.59) | 0.12 | 25.83 (5.59; 49.95) | 0.01 | 36.93 (14.9; 63.2) | <0.01 | 10.92 (-5.26; 29.85) | 0.20 |
| Competition Year | | | | | | | | |
| 2014 | Ref | | Ref | | | | | |
| 2015 + | 7.17 (-12.63; 31.47) | 0.51 | 40.55 (-4.25; 106.31) | 0.08 | -41.04 (-54.1; -24.3) | <0.01 | -13.21 (-36.5; 18.6) | 0.37 |
| **Reviewer Workload** | | | | | | | | |
| # of reviewers per chair | 0.04 (-0.32; 0.41) | 0.81 | 0.09 (-0.48; 0.66) | 0.76 | 0.17 (-0.2; 0.5) | 0.38 | 0.09 (-0.35; 0.53) | 0.69 |
| Mean # applications per reviewer | 4.93 (1.49; 8.48) | <0.01 | 5.05 (-0.68; 11.11) | 0.08 | 3.00 (-2.5; 8.8) | 0.29 | -1.02 (-8.43; 6.98) | 0.80 |
| **Reviewer Characteristics** | | | | | | | | |
| Mean age reviewers | 1.17 (0.41; 1.94) | <0.01 | 0.01 (-1.25; 1.28) | 0.99 | -1.74 (-2.9; -0.6) | <0.01 | -0.01 (-1.18; 1.17) | 0.98 |
| % female reviewers | 0.26 (-15.23; 18.57) | 0.98 | -2.27 (-29.54; 35.57) | 0.89 | 21.63 (-9.6; 63.7) | 0.2 | 30.71 (-8.19; 86.08) | 0.14 |
| Mean years of experience for reviewer | -9.62 (-15.09; -3.80) | <0.01 | 0.62 (-10.56; 13.20) | 0.92 | -4.75 (-13.6; 5.0) | 0.33 | 6.65 (-7.01; 22.32) | 0.36 |
| Pct high expertise reviewers | -14.23 (-27.35; 1.26) | 0.07 | -15.58 (-34.96; 9.59) | 0.20 | -29.14 (-45.3; -8.2) | 0.01 | -35.35 (-53.90; -9.32) | 0.01 |
| % reviewers with applications in same domain | -30.05 (-41.36; -16.55) | <0.01 | -10.69 (-34.38; 21.54) | 0.47 | -15.26 (-36.4; 12.9) | 0.26 | -33.75 (-51.65; -9.22) | 0.01 |
| **Review Process** | | | | | | | | |
| Conflicts on review panel | | | | | | | | |
| None | Ref | | Ref | | Ref | | Ref | |
| At least one | -20.51 (-27.49; -12.86) | <0.01 | 12.54 (-3.47; 31.21) | 0.13 | -26.99 (-48.8; 4.1) | 0.08 | -16.39 (-38.50; 13.68) | 0.25 |
| On-Line Discussion | | | | | | | | |
| No | Ref | | Ref | | Ref | | Ref | |
| Yes | 0.07 (-11.28; 12.88) | 0.99 | 164.15 (124.1; 211.2) | <0.01 | 24.92 (4.3; 49.6) | 0.02 | 96.58 (48.4; 160.4) | <0.01 |

The multiple regression models were specified as follows

Phase 1 & 2: $Y$ (variance rating) $= x_0$ (intercept) $+ x_1$ ($H-index$) $+ x_2$ (PI age) $+ x_3$ (female PI) $+ x_4$ (early career) $+ x_5$ (applied science) $+ x_6$ (2015 competition) $+ x_7$ (reviewers/chair) $+ x_8$ (mean applications per reviewer) $+ x_9$ (mean age reviewers) $+ x_{10}$ (% female reviewers) $+ x_{11}$ (mean reviewer experience) $+ x_{12}$ (% high expertise reviewers) $+ x_{13}$ (% reviewers in same scientific domain) $+ x_{14}$ (conflict on the panel) $+ x_{15}$ (on line discussion) $+ x_{16}$(log n collaborators) $+ x \in$ (residual variance)

Phase 1 & 2: $Y$ (variance ranking) $= x_0$ (intercept) $+ x_1$ ($H-index$) $+ x_2$ (PI age) $+ x_3$ (female PI) $+ x_4$ (early career) $+ x_5$ (applied science) $+ x_6$ (2015 competition) $+ x_7$ (reviewers/chair) $+ x_8$ (mean applications per reviewer) $+ x_9$ (mean age reviewers) $+ x_{10}$ (% female reviewers) $+ x_{11}$ (mean reviewer experience) $+ x_{12}$ (% high expertise reviewers) $+ x_{13}$ (% reviewers in same scientific domain) $+ x_{14}$ (conflict on the panel) $+ x_{15}$ (on line discussion) $+ x_{16}$(log n collaborators) $+ x \in$ (residual variance)

**Table 5. Association between applicant, application and review characteristics and final application score for the two methods: Rating and ranking.**

| | Phase 1 Leadership & Vision | | | | Phase 2 Research Methods | | | |
| | Final Rating | Final Percentile Rank | | | Final Rating | | Final Percentile Rank | |
| | Estimate (95% CI) | P-Value | Estimate (95% CI) | P-Value | Estimate (95% CI) | P-Value | Estimate (95% CI) | P-Value |
|---|---|---|---|---|---|---|---|---|
| **Applicant Characteristics** | | | | | | | | |
| H index | 1.31 (1.12; 1.50) | <0.01 | 2.17 (1.86; 2.48) | <0.01 | 0.20 (0.00; 0.40) | 0.05 | 0.61 (0.10; 1.12) | 0.02 |
| H index * H index | -0.01 (-0.02; -0.01) | <0.01 | -0.02 (-0.03; -0.01) | <0.01 | -0.14 (-0.28; 0.01) | 0.07 | -0.47 (-0.85; -0.08) | 0.02 |
| Total N collaborators (log) | -0.46 (-0.57; 0.34) | <0.011 | -0.87 (-1.07; -0.67) | <0.01 | 0.00 (-0.00; 0.00) | 0.92 | 0.00 (-0.01; 0.01) | 0.88 |
| PI age | 0.14 (0.07; 0.21) | <0.01 | 0.32 (0.19; 0.46) | <0.01 | 0.02 (-0.08; 0.11) | 0.74 | 0.07 (-0.17; 0.31) | 0.58 |
| Sex | | | | | | | | |
| Male | Ref | | Ref | | Ref | | Ref | |
| Female | -1.62 (-2.77; -0.46) | <0.01 | -1.44 (-3.59; 0.72) | 0.199 | -0.17 (-1.74; 1.40) | 0.83 | 1.67 (-2.21; 5.54) | 0.4 |
| Early Career | | | | | | | | |
| No | Ref | | Ref | | Ref | | Ref | |
| Yes | -2.66 (-4.21; -1.11) | <0.01 | -0.04 (-2.82; 2.73) | 0.98 | -3.23 (-5.46; -1.01) | < 0.01 | -7.21 (-13.01; -1.41) | 0.01 |
| **Application Characteristics** | | | | | | | | |
| Scientific Domain | | | | | | | | |
| Basic Science | Ref | | Ref | | Ref | | Ref | |
| Applied Science | -0.36 (-1.57; 0.84) | 0.55 | 0.38 (-2.02; 2.78) | 0.76 | -4.65 (-6.49; -2.81) | < 0.01 | -4.44 (-9.05; 0.17) | 0.06 |
| Competition Year | | | | | | | | |
| 2014 | Ref | | Ref | | Ref | | Ref | |
| 2015 + | -0.11 (-2.19; 1.96) | 0.92 | -11.42 (-15.55; -7.29) | <0.01 | 2.93 (0.49; 5.38) | 0.02 | -7.97 (-14.39; -1.55) | 0.02 |
| **Reviewer Workload** | | | | | | | | |
| # of reviewers per chair | 0.11 (0.07; 0.14) | <0.01 | 0.33 (0.25; 0.40) | <0.01 | 0.03 (0.00; 0.07) | 0.04 | 0.04 (-0.05; 0.13) | 0.34 |
| Mean # applications per reviewer | -0.42 (-0.76; -0.08) | 0.02 | 0.53 (-0.14; 1.19) | 0.12 | -0.24 (-0.82; 0.34) | 0.42 | 0.51 (-0.92; 1.94) | 0.48 |
| **Reviewer Characteristics** | | | | | | | | |
| Mean age reviewers | -0.13 (-0.21; -0.05) | <0.01 | -0.26 (-0.42; -0.09) | <0.01 | 0.16 (0.03; 0.28) | 0.01 | 0.17 (-0.15; 0.50) | 0.29 |
| % female reviewers | 0.59 (-1.27; 2.45) | 0.53 | 1.48 (-2.19; 5.16) | 0.43 | -1.57 (-4.42; 1.29) | 0.28 | -0.54 (-8.02; 6.94) | 0.89 |
| Mean years of experience for reviewer | -0.45 (-1.00; 0.09) | 0.10 | -2.12 (-3.34; -0.89) | <0.01 | 0.34 (-0.50; 1.19) | 0.43 | -1.22 (-3.55; 1.10) | 0.3 |
| Pct high expertise reviewers | 1.45 (-0.11; 3.01) | 0.077 | -0.15 (-3.52; 3.23) | 0.93 | 3.29 (0.94; 5.65) | < 0.01 | 6.17 (-0.09; 12.44) | 0.05 |
| % reviewers with applications in same domain | 0.92 (-0.75; 2.59) | 0.28 | -1.10 (-4.76; 2.56) | 0.55 | 1.97 (-0.74; 4.68) | 0.15 | 2.38 (-4.83; 9.60) | 0.52 |
| **Review Process** | | | | | | | | |
| Conflicts on the review panel | | | | | | | | |
| None | Ref | | Ref | | Ref | | Ref | |
| At least 1 | 6.28 (5.14; 7.41) | <0.01 | 10.69 (8.62; 12.75) | <0.01 | 4.37 (-0.09; 8.84) | 0.06 | 9.19 (-0.96; 19.34) | 0.08 |
| On-Line Discussion | | | | | | | | |
| No | Ref | | Ref | | Ref | | Ref | |

*(Continued)*

**Table 5.** (Continued)

| | Phase 1 Leadership & Vision | | | | Phase 2 Research Methods | | | |
| --- | --- | --- | --- | --- | --- | --- | --- | --- |
| | Final Rating | | Final Percentile Rank | | Final Rating | | Final Percentile Rank | |
| | Estimate (95% CI) | P-Value | Estimate (95% CI) | P-Value | Estimate (95% CI) | P-Value | Estimate (95% CI) | P-Value |
| Yes | 1.74 (0.82; 2.65) | <0.01<0.012 | 2.39 (-0.02; 4.80) | 0.052 | 0.29 (-1.22; 1.79) | 0.71 | 5.76 (1.15; 10.37) | 0.01 |

The multiple regression models were specified as follows

Phase 1 & 2: $Y$ (final rating) $= x0$ (intercept) $+ x1$ ($H - index$) $+ x2$ ($H - eindex*H - index$) $+ x3$(PI age) $+ x4$(female PI) $+ x5$ (early career) $+ x6$ (applied science) $+ x7$ (2015 competition) $+ x8$ (reviewers/chair) $+ x9$ (mean applications per reviewer) $+ x10$ (mean age reviewers) $+ x11$ (% female reviewers) $+ x12$(mean reviewer experience) $+ x13$ (% high expertise reviewers) $+ x14$ (% reviewers in same scientific domain) $+ x15$ (conflict on the panel) $+ x16$ (on line discussion) $+ x17$(log n collaborators) $+ x \in$ (residual variance)

Phase 1 &2: $Y$ (final ranking) $= x0$ (intercept) $+ x1$ ($H - index$) $+ x2$ ($H - eindex*H - index$) $+ x3$(PI age) $+ x4$(female PI) $+ x5$ (early career) $+ x6$ (applied science) $+ x7$ (2015 competition) $+ x8$ (reviewers/chair) $+ x9$ (mean applications per reviewer) $+ x10$ (mean age reviewers) $+ x11$ (% female reviewers) $+ x12$(mean reviewer experience) $+ x13$ (% high expertise reviewers) $+ x14$ (% reviewers in same scientific domain) $+ x15$ (conflict on the panel) $+ x16$ (on line discussion) $+ x17$(log n collaborators) $+ x \in$ (residual variance)

was responsible for was positively associated with higher ratings and ranking of an application in phase 1, and marginally higher ratings but not ranking in phase 2. The number of applications the reviewer was assigned was associated with significantly lower ratings but had no impact on ranking. This association was only evident in phase 1, where the mean number of applications per reviewer was 12.6 compared to 9.5 for phase 2. With respect to reviewer characteristics, having a higher percentage of reviewers with high expertise was associated with higher scores and ranks in phase 2. In contrast, reviewers whose applications were in the same domain as the applicant did not influence application score or rank. Although the sex mix of the reviewers did not significantly influence application score or rank, older reviewers were

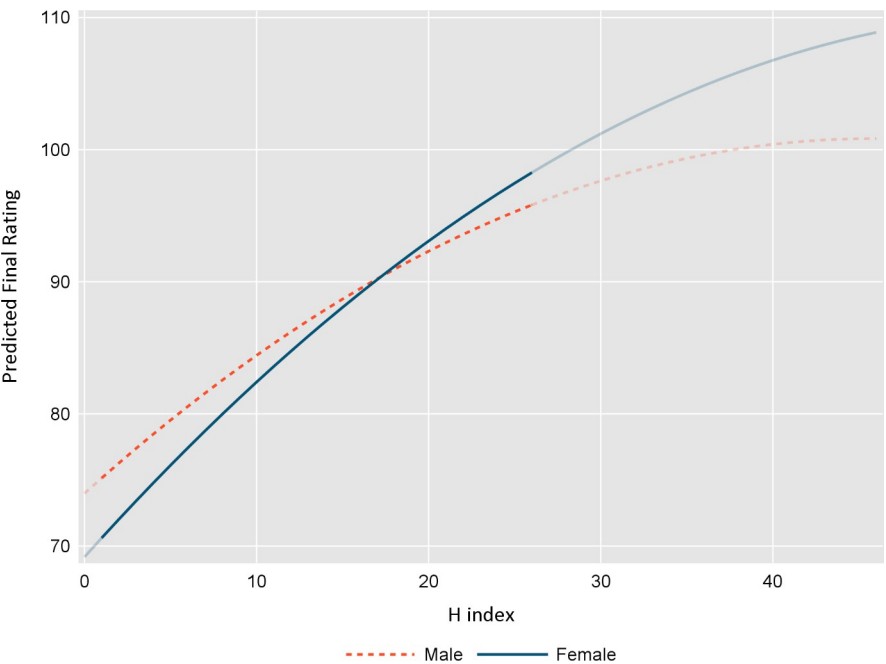

**Fig 1. Association between H-index and final application rating, by sex.**

more likely to score and rank an application lower than younger reviewers in phase 1, as were reviewers with more experience. Finally, on-line discussion of an application was associated with higher scores and ranks in phase 1 and 2. Of note, application scores and ranks were higher in both phase 1 and 2 when one or more invited reviewers declared a conflict, although only significant for phase 1.

## Discussion

This study provided one of the first opportunities to compare the reliability and potential sources of bias of two methods of peer review scoring—rating and ranking—in a national funding competition at the Canadian Institutes of Health Research. We found that ranking was more reliable than rating, and less susceptible to characteristics of the review panel such as level of expertise and experience for both reliability and potential sources of bias. However, both rating and ranking penalized early career investigators and favoured older applicants. Sex bias was evident only for rating and when the applicant's H-index was at the lower end of the H-index distribution.

Theoretically, the reliability of rating would be more likely influenced by reviewer experience than that of ranking, as more experienced reviewers would have a more stable conceptual model of the "ideal application" against which an application could be judged [20]. Our findings are consistent with this hypothesis. Greater reviewer experience was positively associated with smaller rater variance for rating, but had no influence on ranking. Of note, rater variance was also smaller when there was less uncertainty of an applicant's track record, with lower levels of rater variance for more senior investigators and those with a higher H-index. Rater variance was significantly higher for early career investigators, although these effects were less pronounced for ranking compared to rating. Consistent with prior research [9, 18, 21, 36], reliability was higher for assessment of the researcher (phase 1), than for the research project (phase 2) for both rating and ranking. In this study, lower reliabilities for the research project may also be related to the selection of only the best applications to proceed to phase 2. Even though applications were rated by an independent set of reviewers in phase 2, using a different set of criteria, the quality of the candidates and applications selected to phase 2 may be more homogenious than in phase 1 making it more difficult for reviewers to distinguish amongst higher quality proposals. In this context, ranking outperformed rating, likely because it forced reviewers to distinguish among applications given the same score by breaking ties among applications with the same absolute score. This would effectively increase the between application variance relative to the within rater variance thereby increasing reliability. The mean number of applications with ties was substantial in both phase 1 and 2 and most reviewers had at least one tie that needed to be broken in both phases. The exercise of tie breaking likely allowed reviewers to reflect on more nuanced differences between applications, and possibly to correct for more extreme high or low ratings of some applicants.

As the majority of funding agencies use peer review rating to assess the quality of an application and determine funding, the reliability of peer review should be improved by using a ranking rather than a rating system. Ranking would also provide greater flexibility in recruiting reviewers from various disciplines with different levels of experience without compromising reliability. Ranking may also confer reductions in potential sources of bias in application assessment. In our study ranking compared to rating significantly reduced the negative bias in scoring for female and early career applicants.

In this study, the reliability of ranking could have been improved if reviewers had been given an equivalent number of applications to review. To adjust for inequities in the number of applications reviewed, ranks were converted to percentile ranks, which added some "noise"

to the application score, reducing the ICC based on phase 1 ranking from 0.6 to 0.59. For example, a ranking of second best for a reviewer who had 5 applications to review and rank would result in a percentile rank of 80%, whereas a rank of second for a reviewer who had 12 applications would result in a percentile rank of 91.7%. Another factor that may have contributed to random measurement error in ranking is an unequal distribution of the quality of applications assigned to each reviewer, a challenge that should be addressed in future research of peer-review ranking methods.

Similar to other studies [5], we found that older applicants received higher scores and ranks compared to younger applicants with the same H-index. Also, extending to others' findings [37], we found female applicants were more likely to receive lower scores and ranks on their applications. Our analysis provides some insight into the reason for these differences, as they pertain only to male and female applicants with the same H-index but at the lower end of the H-index distribution. At higher values of the H-index, female applicants perform the same or better than males in terms of final application ranks and scores. The policy implications of these findings are important, as these unconscious biases in rating will influence those at early career stages, potentially discouraging female applicants from continuing to pursue a scientific career. The Matthew effect, where early failure in application for research funding among equivalent candidates discourages subsequent efforts for funding, may be one of the reasons why female scientists are less likely to progress to more senior positions in their research career [24, 38, 39]. Indeed, many countries have noted an over-representation of males in senior academic and scientific positions and awards, leading to proactive policies to support equity, diversity and inclusion in all areas of science [40–42].

With respect to the review process, we found that on-line discussion of an application was associated with higher rater variance, which may be expected, as efforts would be selectively made to resolve differences in review opinion among reviewers with very divergent scores and ranks. Of interest, on-line discussion had a positive effect, significantly increasing the scores and ranks for applications that were discussed on-line compared to those that were not. The other interesting finding was the effect of conflicts. In prior research, we and others have shown that conflicts on a panel with an application are associated with higher scores [5, 11, 24, 34, 35], even when those in conflict do not participate in scoring or the discussion. It has been assumed that this phenomenon is related to cronyism, the influence of professional networks or group dynamics within the peer review meetings [5, 24, 34, 35]. We also note the same positive effects of conflicts on scores, but as reviews were done asynchronously and virtually, and no reviewer in conflict was involved in scoring or discussion, these prior theories for the positive effects of conflicts are not supported. We speculate that applications where there are reviewer conflicts are being submitted by scientists in highly productive groups or networks, and if this is the case, conflicts may be a marker of the quality of an application. This hypothesis needs to be evaluated in future research.

There are important limitations to consider in interpreting our findings. First, we had no direct measurement of the quality of an application. We used the principal applicant's H-index as a proxy for application quality, and while it was strongly associated with the final application score and rank, we have no direct measure if its validity. Second, we do not know what the impact is of rating an application first and then ranking, as opposed to ranking and then rating. We suspect that we may have underestimated the benefits of ranking, and future research should address this potential bias. Finally, we had no information about other applicant characteristics, such as race or disability, which have been associated with bias in assessment [29, 43].

In conclusion, ranking compared to rating provides a more reliable assessment of the quality of research applications, was not as influenced by reviewer expertise or experience, and was associated with fewer potential sources of bias.

## Author Contributions

**Conceptualization:** Robyn Tamblyn, Nadyne Girard, James Hanley, Bettina Habib, Adrian Mota, Karim M. Khan, Clare L. Ardern.

**Data curation:** Nadyne Girard.

**Formal analysis:** Nadyne Girard.

**Funding acquisition:** Robyn Tamblyn.

**Investigation:** Robyn Tamblyn.

**Methodology:** Robyn Tamblyn, Nadyne Girard, James Hanley, Adrian Mota, Karim M. Khan, Clare L. Ardern.

**Project administration:** Bettina Habib.

**Resources:** Robyn Tamblyn.

**Supervision:** Robyn Tamblyn.

**Validation:** Nadyne Girard, Bettina Habib.

**Visualization:** Nadyne Girard.

**Writing – original draft:** Robyn Tamblyn.

**Writing – review & editing:** Robyn Tamblyn, Nadyne Girard, James Hanley, Bettina Habib, Adrian Mota, Karim M. Khan, Clare L. Ardern.

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
