## [Decision Letter · Decision Letter 0]

1 Nov 2022

PONE-D-22-19788Ranking versus Rating in Peer Review of Research Grant ApplicationsPLOS ONE

Dear Dr. Tamblyn,

Thank you for submitting your manuscript to PLOS ONE. After careful consideration, we feel that it has merit but does not fully meet PLOS ONE’s publication criteria as it currently stands. Therefore, we invite you to submit a revised version of the manuscript that addresses the points raised during the review process. The reviewer raises concerns about the soundness of the statistical analysis that need to be addressed.

We look forward to receiving your revised manuscript.

Kind regards,

Luís A. Nunes Amaral, Ph.D.

Academic Editor

PLOS ONE

Journal Requirements:

2. Please change "female” or "male" to "woman” or "man" as appropriate, when used as a noun (see for instance https://apastyle.apa.org/style-grammar-guidelines/bias-free-language/gender).

Reviewers' comments:

Reviewer's Responses to Questions

**Comments to the Author**

1. Is the manuscript technically sound, and do the data support the conclusions?

Reviewer #1: Partly

2. Has the statistical analysis been performed appropriately and rigorously? 

Reviewer #1: No

3. Have the authors made all data underlying the findings in their manuscript fully available?

Reviewer #1: No

4. Is the manuscript presented in an intelligible fashion and written in standard English?

Reviewer #1: No

5. Review Comments to the Author

Reviewer #1: In this manuscript, Tamblyn et al. analyzed an interesting dataset about grant peer-review in Canada. The main purpose of the analysis is to compare the reliability and bias of peer-review under two different scenarios: ranking versus rating. The main conclusion is that ranking seems to increase reliability while reducing bias. Furthermore, the authors also found some factors correlated with variance and outcome of the review.

My comments are listed below:

Comments about conceptual issues:

1. Perhaps the most convincing result in the paper is that ranking increases the reliability of the review compared to raw scores (Table 3). However, the mechanism articulated by the authors is a bit vague to me. The authors claimed that the benefit of ranking stems from removing the requirement for a stable internalized standard, which is difficult for inexperienced reviewers compared to experienced reviewers. This seems an overly complicated explanation. A more straightforward explanation would be that ranking controls for leniency (1) variation among evaluators, so the evaluation is more robust. Either way, I recommend a more thorough discussion of potential mechanisms and perhaps lay out future tests that can distinguish between these hypotheses.

2. Compared to the results about reliability, the results about reducing bias (table 5) are much less convincing. The main reason is that the quality proxy for proposals is H-index. While this might be partially justified in the first stage evaluation (since the first stage appears to be evaluating people, not the grant proposal), it is important to acknowledge that H-index can be a signal of prestige instead of researcher quality, so the results in table 5 might as well reflect that ranking and rating weighted different kinds of bias differently;

3. Given that the authors found relatively convincing evidence of ranking reducing reliability versus reducing bias, it might be beneficial to clearly contrast these two concepts in the introduction, as many readers can lump the two concepts together and believe they are concordant measures of peer-review quality. Sometimes, larger variance in evaluation can be good for better evaluation outcomes since pooling different opinions can result in the ‘wisdom of the crowd (2,3)’;

Comments about statistics:

1. I have a minor concern about ICC measures for reliability (I might be wrong since I am not an expert on this measure). If I remember correctly, the ICC requires some normality assumption (4). This is likely true for raw rating score but might not be valid for percentile ranking, as one would expect the percentile ranking would follow a uniform distribution between 0-1. Or maybe it will be approximately normal after taking the mean?

2. If I understand correctly, the regressions in Tables 4 and 5 are also used to search for factors that are significantly associated with variance (another way of measuring reliability) and outcome in different stages and schemes. If this is the case, the authors are effectively conducting multiple testing. In this case, a good practice is to perform multiple testing corrections, even for multiple regression (5). At the very least, authors can display the p-values with higher precision so that interested readers can gauge the results when needed. Along the same lines, if figure 1 shows the only significant interaction terms, then the (adjusted) p-value of the interaction term should be demonstrated, as figure 1 is what remains among the several interaction terms tested as described in the main text. Besides, to show the non-linear effect in figure 1, confidence intervals should be plotted;

Comment about presentation:

1. Finally, the organization of the paper should be substantially improved for readability. For example, a schematic figure showing the reviewing process of the CIHR program would be helpful. Furthermore, the formula for multiple regressions conducted should be explicitly shown instead of relying on a verbal description. In addition, the results would be much easier to read if divided into different sections (with titles) according to the conclusions.

Overall, I recommend minor revisions before further consideration.

References:

(1) Sampat, B., & Williams, H. L. (2019). How do patents affect follow-on innovation? Evidence from the human genome. American Economic Review, 109(1), 203-36.

(2) Shi, F., Teplitskiy, M., Duede, E., & Evans, J. A. (2019). The wisdom of polarized crowds. Nature human behaviour, 3(4), 329-336.

(3) Sun, M., Barry Danfa, J., & Teplitskiy, M. (2022). Does double-blind peer review reduce bias? Evidence from a top computer science conference. Journal of the Association for Information Science and Technology, 73(6), 811-819.

(4) Nakagawa, S., & Schielzeth, H. (2010). Repeatability for Gaussian and non-Gaussian data: a practical guide for biologists. Biological Reviews, 85(4), 935-956.

(5) Perrett, D. J. M. J. J., Schaffer, J., Piccone, A., & Roozeboom, M. (2006). Bonferroni adjustments in tests for regression coefficients. Multiple Linear Regression Viewpoints, 32(1), 1-6.

6. PLOS authors have the option to publish the peer review history of their article (what does this mean?). If published, this will include your full peer review and any attached files.

Reviewer #1: No

---

## [Author Response · Author response to Decision Letter 0]

5 Dec 2022

We have included a detailed response to the reviewer's comments in the attached files. 

We have also amended the data availability statement and have modified the formatting of the manuscript (final version only, not tracked version) to fit the style requirements of PLOS ONE.

---

## [Decision Letter · Decision Letter 1]

10 Apr 2023

PONE-D-22-19788R1Ranking versus rating in peer review of research grant applicationsPLOS ONE

Dear Dr. Tamblyn,

Thank you for submitting your manuscript to PLOS ONE. After careful consideration, we feel that it has merit but does not fully meet PLOS ONE’s publication criteria as it currently stands. Therefore, we invite you to submit a revised version of the manuscript that addresses the points raised during the review process.

See comments below. 

We look forward to receiving your revised manuscript.

Kind regards,

Julian D. Cortes

Academic Editor

PLOS ONE

**Additional Editor Comments:**

Dear author/s, thanks for submitting your work to PLoS ONE,

I contrasted the assessment of two reviewers of your work. Considering that the overall assessment resulted in a major and a minor revision, the article still needs further adjustments, particularly in the literature review which will enrich the discussion of the findings, and the methodology/statistical method applied.

Contrasting reviewers’ assessment with PLoS ONE’s requisites for publication, the article should be strengthened in the following terms:

• Experiments, statistics, and other analyzes are performed to a high technical standard and are described in sufficient detail.

• Conclusions are presented appropriately and are supported by the data.

I hope you can incorporate the above suggestions to improve your already valuable work.

Sincerely,

Julián D. Cortés

Associate Editor

Reviewers' comments:

Reviewer's Responses to Questions

**Comments to the Author**

1. If the authors have adequately addressed your comments raised in a previous round of review and you feel that this manuscript is now acceptable for publication, you may indicate that here to bypass the “Comments to the Author” section, enter your conflict of interest statement in the “Confidential to Editor” section, and submit your "Accept" recommendation.

Reviewer #1: All comments have been addressed

Reviewer #2: (No Response)

Reviewer #3: (No Response)

2. Is the manuscript technically sound, and do the data support the conclusions?

Reviewer #1: Yes

Reviewer #2: Yes

Reviewer #3: Partly

3. Has the statistical analysis been performed appropriately and rigorously? 

Reviewer #1: Yes

Reviewer #2: Yes

Reviewer #3: No

4. Have the authors made all data underlying the findings in their manuscript fully available?

Reviewer #1: No

Reviewer #2: No

Reviewer #3: No

5. Is the manuscript presented in an intelligible fashion and written in standard English?

Reviewer #1: Yes

Reviewer #2: Yes

Reviewer #3: Yes

6. Review Comments to the Author

Reviewer #1: The authors have addressed all my comments. I have no further questions. The only suggestion is that the formula presentation could possibly be improved via LaTex. But I assume this can be done in the final proofreading process.

Reviewer #2: The authors suggest that the peer review process used in determining the funding priority of submitted proposals has been plagued by reports of poor reliability and that one potential solution may lie in examining other schemas of evaluation, including a ranking process, which may be more reliable and may be less prone to bias. To examine this, the authors compare the reliability associated with application rating with those of application ranking in 3,156 applications that have been submitted to the Canadian Institutes of Health Research funding agency. In general, the authors found that ranking was more reliable than

rating and “less susceptible to the characteristics of the review panel, such as level of

expertise and experience, for both reliability and potential sources of bias.” This natural experiment is important and the authors should be lauded for taking this analysis on, as access to these types of data are limited. In general, this is an important area, and improving upon current review processes is important not only to provide more consistent output but for the credibility of the scientific process itself.

First, the differences observed in the reliability measures for rating and ranking are an important finding. While it seems the rating and ranking are independent processes, they are also highly correlated. Is there an assumption here that reviewers use the same criteria for ranking and rating? If they are different, do the authors have any insight into the ways they diverge? It may be helpful to add a little language in the discussion addressing what the authors think might be the differences in the decision-making processes peer reviewers use for rating vs ranking?

Also, the reliabilities for rating and ranking are somewhat similar for stage 1 and better than that of stage 2 (possibly because reviewers in stage 1 are mostly discriminating between good versus bad proposals?). For stage 2, reliability for both rating and ranking is worse, but the difference between them has grown, suggesting ranking is more clearly advantageous over rating for stage 2 (perhaps because judgements at this stage are more difficult; i.e. it is more difficult to discriminate between good and great applications via a rating mechanism?). Do the authors feel the utility of ranking over rating improves for proposals that are in the good/great zone? This is important as this may be a range where peer review is the least reliable or at least has the most difficulty in discriminating between proposals and may be most prone to subjectivity and bias.

Another area that might be undersold here a bit is the inherent power of ranking for tie-breaking. Could the authors elaborate on this in the analysis (e.g. how many rating ties are broken by the ranking process?).

While a good portion of the analysis is spent on bias, it is not entirely clear that ranking is devoid of bias or if it is less susceptible to bias, why that is in the context of peer review? Maybe the authors might add a little language in the discussion to address this.

A minor concern, the authors state that “Ranking would also provide greater flexibility in recruiting reviewers with different levels of experience and expertise without compromising reliability.” But is the goal to reduce expertise levels of reviewers?

Also, the authors mention the online discussion increased ranking variance “With respect to the review process, we found that on-line discussion of an application was associated with higher rater variance, which may be expected, as efforts would be selectively made to resolve differences in review opinion among reviewers with very divergent scores” Maybe the authors could provide more description about how proposals were selected for online discussion (I’m presuming there was a cut-off for discrepant scores?). If one looked at scores before and after discussion, would you would see a resolving of scores, i.e. scores coming together (reduced variance)?

Reviewer #3: The authors present the work titled: "Ranking versus rating in peer review of research grant applications." Though the topic is of great relevance, it has several flaws that authors should address to enhance the quality of the manuscript.

First, the literature review omits several relevant aspects that the reader should know to understand the full scope and implications of the research. Note, however, that the literature on grant funding evaluations is enormous. A potential contribution in this regard relates to a comprehensive literature review. Unfortunately, the literature review provided by the authors is weak. The authors could have included several other factors to illustrate or explain their empirical strategy. For example, the Matthew effect (Bol et al, 2018), gender effects (Eloy et al, 2013), the use of multi-criteria approaches (Oztaysi et al, 2017), the effects per discipline (Jerrim & Vries, 2020); implementation considerations (Neta et al 2015), or previous recommendations for reliability and validity (Marsh et al 2008) are just a few points of an extensive list of missed points.

Another concern is the statistical treatment. In particular, the authors have missed potential problems associated with endogeneity in the data. Endogeneity broadly refers to situations in which an explanatory variable (e.g., the H-Index of the applicant) correlates with the error term of the regression equation (tables 4 and 5). In this case, if authors do not control for how many co-authors an applicant has worked with in previous works, this might increase the error term of the regression equation because H-index correlates with the number of co-authors (Vinkler, 2023).

Due to the issues described above, the manuscript can not be accepted in its current form and the authors should address them before an additional consideration for the study.

SUGGESTED REFERENCES

Bol, T., de Vaan, M., & van de Rijt, A. (2018). The Matthew effect in science funding. Proceedings of the National Academy of Sciences, 115(19), 4887-4890.

Eloy, J. A., Svider, P. F., Kovalerchik, O., Baredes, S., Kalyoussef, E., & Chandrasekhar, S. S. (2013). Gender differences in successful NIH grant funding in otolaryngology. Otolaryngology--Head and Neck Surgery, 149(1), 77-83.

Jerrim, J., & Vries, R. D. (2020). Are peer-reviews of grant proposals reliable? An analysis of Economic and Social Research Council (ESRC) funding applications. The Social Science Journal, 1-19.

Marsh, H. W., Jayasinghe, U. W., & Bond, N. W. (2008). Improving the peer-review process for grant applications: reliability, validity, bias, and generalizability. American psychologist, 63(3), 160.

Neta, G., Sanchez, M. A., Chambers, D. A., Phillips, S. M., Leyva, B., Cynkin, L., ... & Vinson, C. (2015). Implementation science in cancer prevention and control: a decade of grant funding by the National Cancer Institute and future directions. Implementation Science, 10, 1-10.

Oztaysi, B., Onar, S. C., Goztepe, K., & Kahraman, C. (2017). Evaluation of research proposals for grant funding using interval-valued intuitionistic fuzzy sets. Soft Computing, 21, 1203-1218.

Vinkler, P. (2023). Impact of the number and rank of coauthors on h-index and π-index. The part-impact method. Scientometrics, 1-21.

7. PLOS authors have the option to publish the peer review history of their article (what does this mean?). If published, this will include your full peer review and any attached files.

Reviewer #1: No

Reviewer #2: No

Reviewer #3: **Yes: **Juan Carlos Correa Nuñez

---

## [Author Response · Author response to Decision Letter 1]

8 May 2023

Responses below have also been appended as an attached Word document. 

Reviewer #1: The authors have addressed all my comments. I have no further questions. The only suggestion is that the formula presentation could possibly be improved via LaTex. But I assume this can be done in the final proofreading process.

 The equations in the table footnotes have been re-formatted in LaTex

Reviewer #2: The authors suggest that the peer review process used in determining the funding priority of submitted proposals has been plagued by reports of poor reliability and that one potential solution may lie in examining other schemas of evaluation, including a ranking process, which may be more reliable and may be less prone to bias. To examine this, the authors compare the reliability associated with application rating with those of application ranking in 3,156 applications that have been submitted to the Canadian Institutes of Health Research funding agency. In general, the authors found that ranking was more reliable than rating and “less susceptible to the characteristics of the review panel, such as level of expertise and experience, for both reliability and potential sources of bias.” This natural experiment is important and the authors should be lauded for taking this analysis on, as access to these types of data are limited. In general, this is an important area, and improving upon current review processes is important not only to provide more consistent output but for the credibility of the scientific process itself.

 Thank-you for your comments and support for this work. 

First, the differences observed in the reliability measures for rating and ranking are an important finding. While it seems the rating and ranking are independent processes, they are also highly correlated. Is there an assumption here that reviewers use the same criteria for ranking and rating? If they are different, do the authors have any insight into the ways they diverge? It may be helpful to add a little language in the discussion addressing what the authors think might be the differences in the decision-making processes peer reviewers use for rating vs ranking?

 The advantage of this natural experiment was that the criteria used to rate and rank applications were the same. While we have no information on the decision-making process of reviewers, we have hypothesized possible mechanisms that could be operating in the ranking process that would lead to improved reliability and reduction in bias to the discussion. 

Also, the reliabilities for rating and ranking are somewhat similar for stage 1 and better than that of stage 2 (possibly because reviewers in stage 1 are mostly discriminating between good versus bad proposals?). For stage 2, reliability for both rating and ranking is worse, but the difference between them has grown, suggesting ranking is more clearly advantageous over rating for stage 2 (perhaps because judgements at this stage are more difficult; i.e. it is more difficult to discriminate between good and great applications via a rating mechanism?). Do the authors feel the utility of ranking over rating improves for proposals that are in the good/great zone? This is important as this may be a range where peer review is the least reliable or at least has the most difficulty in discriminating between proposals and may be most prone to subjectivity and bias.

Another area that might be undersold here a bit is the inherent power of ranking for tie-breaking. Could the authors elaborate on this in the analysis (e.g. how many rating ties are broken by the ranking process?).

 This is an excellent point and we have added this information on tie breaking to the results reported relative to Table 3. As the reviewer suspects, tie breaking was a frequent occurrence. In phase 1, the average number of ties that needed to be broken by a reviewer were for 31.7% of their application reviews and only 16% of reviewers had no ties that needed to be broken. In phase 2,ties needed to eb broken in 28.3% of applications and by 75.4% of reviewers. 

While a good portion of the analysis is spent on bias, it is not entirely clear that ranking is devoid of bias or if it is less susceptible to bias, why that is in the context of peer review? Maybe the authors might add a little language in the discussion to address this.

 We agree. In our discussion we indicated that ranking may confer reductions in potential sources of bias, we clarified that this comment referred to our findings related to applicant gender and early career status where there was an appreciable difference in the performance of rating and ranking.

A minor concern, the authors state that “Ranking would also provide greater flexibility in recruiting reviewers with different levels of experience and expertise without compromising reliability.” But is the goal to reduce expertise levels of reviewers?

 We realize this sentence is misleading and have revised it as follows..

“Ranking would also provide greater flexibility in recruiting reviewers from various disciplines with different levels of experience without compromising reliability”

Also, the authors mention the online discussion increased ranking variance “With respect to the review process, we found that on-line discussion of an application was associated with higher rater variance, which may be expected, as efforts would be selectively made to resolve differences in review opinion among reviewers with very divergent scores” Maybe the authors could provide more description about how proposals were selected for online discussion (I’m presuming there was a cut-off for discrepant scores?). If one looked at scores before and after discussion, would you would see a resolving of scores, i.e. scores coming together (reduced variance)?

 In the Methods section, we have augmented the description of the peer review process to indicate how applicants were selected for on-line discussion. The reviewer has raised an important point about the potential consequences of discussion on rater variance. Unfortunately only the final rating and ranking were retained by the peer review system so this question cannot be addressed.

Reviewer #3: The authors present the work titled: "Ranking versus rating in peer review of research grant applications." Though the topic is of great relevance, it has several flaws that authors should address to enhance the quality of the manuscript.

First, the literature review omits several relevant aspects that the reader should know to understand the full scope and implications of the research. Note, however, that the literature on grant funding evaluations is enormous. A potential contribution in this regard relates to a comprehensive literature review. Unfortunately, the literature review provided by the authors is weak. The authors could have included several other factors to illustrate or explain their empirical strategy. For example, the Matthew effect (Bol et al, 2018), gender effects (Eloy et al, 2013), the use of multi-criteria approaches (Oztaysi et al, 2017), the effects per discipline (Jerrim & Vries, 2020); implementation considerations (Neta et al 2015), or previous recommendations for reliability and validity (Marsh et al 2008) are just a few points of an extensive list of missed points.

 We thank the reviewer for identifying additional references that were not identified in our literature search, many of which were relevant to the research question addressed in this manuscript. The literature review outlined in the introduction was related to the specific question of reliability and bias in grant applications using two methods, rating and ranking. We cover both the theoretical literature on these forms of assessment, and the results of applications in other contexts as well as in grant review. A comprehensive literature review on all that is known about grant peer review was beyond the scope or intent of this manuscript. Two comprehensive reviews have already been published on this topic, both of which are cited in this manuscript (Guthrie, 2018; Shepherd, 2018). 

Another concern is the statistical treatment. In particular, the authors have missed potential problems associated with endogeneity in the data. Endogeneity broadly refers to situations in which an explanatory variable (e.g., the H-Index of the applicant) correlates with the error term of the regression equation (tables 4 and 5). In this case, if authors do not control for how many co-authors an applicant has worked with in previous works, this might increase the error term of the regression equation because H-index correlates with the number of co-authors (Vinkler, 2023).

 This is a very good point. We have redone the analysis to include the number of unique collaborators listed in the applicant’s publications to address this problem of unmeasured covariates in the error term. The correlation between the H index and number of collaborators is reasonably high (r=0.5). Both the H index and number of collaborators are significantly associated with rater variance and the score in the same direction. However, there is no impact of including number of collaborators on estimates of applicant, reviewer and review characteristics on variance and bias. The new results are presented in Table 4 and 5.

---

## [Decision Letter · Decision Letter 2]

26 Jul 2023

PONE-D-22-19788R2Ranking versus rating in peer review of research grant applicationsPLOS ONE

Dear Dr. Tamblyn,

Thank you for submitting your manuscript to PLOS ONE. After careful consideration, we feel that it has merit but does not fully meet PLOS ONE’s publication criteria as it currently stands. Therefore, we invite you to submit a revised version of the manuscript that addresses the points raised during the review process.

Dear authors,

I’ve been handling your manuscript over the last four months after receiving an editor transfer request. I’m aware of the more than a year reviewing process and, therefore, admire your commitment during this year of revisions and adjustment to your article, following reviewers’ minor and major suggestions.

The first round of revisions received minor and overall favorable concepts. However, in the process of seeking new reviewers to check upon those revisions, before the last round the article received a “major revision,” that you already completed, and in the last round, unfortunately, the article received a “reject” concept.

To assume a nuanced position not only based on the most recent reviews but also those of the first versions of the manuscript managed by the previous editor, I will consider that you could address the concerns of the reviewer below. Mind that there are multiple methodological considerations that might require running analyses and interpret and discuss results with the literature suggested by the reviewer.

I look forward to your revised manuscript. 

We look forward to receiving your revised manuscript.

Kind regards,

Julian D. Cortes

Academic Editor

PLOS ONE

Reviewers' comments:

Reviewer's Responses to Questions

**Comments to the Author**

1. If the authors have adequately addressed your comments raised in a previous round of review and you feel that this manuscript is now acceptable for publication, you may indicate that here to bypass the “Comments to the Author” section, enter your conflict of interest statement in the “Confidential to Editor” section, and submit your "Accept" recommendation.

Reviewer #4: (No Response)

2. Is the manuscript technically sound, and do the data support the conclusions?

Reviewer #4: No

3. Has the statistical analysis been performed appropriately and rigorously? 

Reviewer #4: No

4. Have the authors made all data underlying the findings in their manuscript fully available?

Reviewer #4: No

5. Is the manuscript presented in an intelligible fashion and written in standard English?

Reviewer #4: Yes

6. Review Comments to the Author

Reviewer #4: Based on the title and abstract, this paper appears to be aimed at studying ranking and rating in grant peer review. The authors use a dataset from the Canadian Institutes of Health Research to compare their ranking and rating procedures for grant proposal peer review quality assessement. The main conclusion of the paper is that funding agencies should consider adopting ranking methods to improve the quality of funding decisions in health research.

The authors consider an important problem and attempt to study it using a large grant peer review data set from a reputable funding agency. The question of comparative reliability in ranking and rating is interesting. However, the actual quality assessment procedure that is being studied relative to standard rating is not ranking; it could be more accurately described as “tie-breaking score-based rank percentile” procedure. Therefore, the manuscript’s title and abstract claims are misleading.

Is the manuscript technically sound, and do the data support the conclusions?

There are substantial technical concerns with the manuscript that do not allow one to determine if the data support the conclusions about the use of ranking in grant peer review. The most important concerns include:

1. Although the title and abstract state that reliability of ratings and rankings are compared in the manuscript, that is not the case. Instead, the authors compare reliability of ratings and rank percentiles, obtained after reviewers were asked to break ties in their ratings. Rank percentiles are substantially different from rankings and have distinct properties. Many desirable properties of rankings, such as the direct comparison of proposals without reference to a scale, are lost when converting ranks to rank percentiles.

2. Relatedly, some conclusions noted by the authors are the direct mathematical result of the construction of their “ranking” data. Such conclusions include: (1) the authors note on line 240 that the mean rank percentile is 50%, which is a consequence of their data conversion that would hold for any data source. (2) Similarly, the lack of statistical significance for coefficients in the rank percentile model is likely the result of the “zero-sum” construction of rank percentiles, and not necessarily a true lack of statistical significance.

3. Although authors note that “inclusion of only top scoring applications in phase 2 reduced the reliability of assessment”, they still make the conclusion on lines 356-360 that IRR was found to be smaller for the assessment of research project quality (phase 2) than that of researchers (phase 1). This comparison is flawed. That IRR decreased in phase 2 is primarily a mathematical result of removing the estimated “worst” proposals (see Erosheva, Martinkova, and Lee, 2021).

4. On lines 138-139, the authors note “it was assumed that the distribution of the quality of applications assigned to each reviewer would be equivalent.” What is precisely meant by the phrase “the distribution of the quality of applications…would be equivalent”? Whatever the formal description of this assumption might be, it appears to be quite stringent and unlikely to hold, especially if there was substantial variability in the number of proposals assessed by each reviewer. The authors report the mean numbers of 12.6 and 9.5 of proposals assessed by each reviewer in phases 1 and 2, respectively, but do not comment on the range. Given this implausible assumption, the authors should explore its influence on the results.

5. Comparing the reliability (variance) of ratings and rankings is ultimately challenging given the ordinal properties of rankings. The authors’ attempts to make this comparison, while commendable in goal, are not statistically appropriate.

6. On numerous instances, we believe the authors use the phrase “percent rank” when referring to “rank percentile”. These are different concepts.

7. Interpretation of regression coefficients should be done conditionally on all other variables being held constant.

References:

Erosheva, E. A., Martinková, P., & Lee, C. J. (2021). When zero may not be zero: A cautionary note on the use of inter‐rater reliability in evaluating grant peer review. Journal of the Royal Statistical Society: Series A (Statistics in Society)

Has the statistical analysis been performed appropriately and rigorously?

Please refer to the concerns raised in answering the previous question.

Have the authors made all data underlying the findings in their manuscript fully available?

The data will not be publicly available citing privacy concerns. We suggest the authors attempt to release a de-identified subset of the data publicly, as has been done with similar data from the Swiss NSF in Heyard et al. (2021), the National Institutes of Health in Erosheva et al. (2020), and the American Institute of Biological Sciences in Pearce and Erosheva (2022).

References:

Heyard, R., Ott, M., Salanti, G., & Egger, M. (2022). Rethinking the funding line at the Swiss national science foundation: Bayesian ranking and lottery. Statistics and Public Policy, 9(1), 110-121.

Erosheva, E. A., Grant, S., Chen, M. C., Lindner, M. D., Nakamura, R. K., & Lee, C. J. (2020). NIH peer review: Criterion scores completely account for racial disparities in overall impact scores. Science Advances, 6(23), eaaz4868.

Pearce, M., & Erosheva, E. A. (2022). A unified statistical learning model for rankings and scores with application to grant panel review. Journal of Machine Learning Research, 23(210), 1-33.

Is the manuscript presented in an intelligible fashion and written in standard English?

Yes.

7. PLOS authors have the option to publish the peer review history of their article (what does this mean?). If published, this will include your full peer review and any attached files.

Reviewer #4: No

---

## [Author Response · Author response to Decision Letter 2]

29 Aug 2023

Dear Editor,

Thank you for providing us with the opportunity to respond to comments from a new reviewer of our manuscript. If I am to understand correctly, our responses to the previous sets of reviewers were acceptable but the most recent reviewer has raised new issues not previously identified by the prior sets of reviewers.

Unfortunately, the critique provided by this reviewer was based on a series of assumptions that are incorrect. They include the following:

1. The reviewer has assumed that the unit of analysis is the reviewer (comment 1,2). This is incorrect. The unit of analysis is the application. We measured the characteristics of the application and its score and percentile rank based on the mean of ratings and percentile ranks provided by all reviewers of the application.

2. The reviewer has assumed that there is a range restriction in scores in phase 2, as was the case in the Erosheva paper that was cited by the reviewer (comment 3). This is incorrect. The applications in phase 2 were scored again, using a different set of reviewers and criteria to assess the quality of the research program using the the full score range for their assessment. Our analysis is based on ALL applications, both those that were funded and those that were unsuccessful in both Phase 1 and Phase 2. 

3. The reviewer assumed that the data were ordinal, likely because of confusion about the unit of analysis (comment 5). This is incorrect. The data are continuous as evidenced by the histogram and rug plot shown in response to this comment. Each application received a rating allowing one decimal place of a scale of 0-112 for phase 1 and 0-140 for phase 2 and a ranking from each of the 5 reviewers which was converted by CIHR to a percentile ranking to adjust for differences in the number of applications rated by each reviewer. The arithmetic mean of these values became the application score and percentile rank. 

4. The reviewer assumed that only bivariate analysis was conducted (comment 7). This is incorrect. In the analysis section we indicated that multivariate regression was used to estimate the associations between application, applicant and reviewer characteristics and each outcome, and provided the regression models in the footnotes of each table. By definition, multivariate models provide estimates of the independent association of a given variable, controlling for all other variables in the model. 

Our possibly unclear description of the methodology may have led, in part, to this series of incorrect assumptions. Therefore, we have revised the methods section of the paper to improve clarity and avoid these misunderstandings by future readers.

Our detailed responses to each comment follow. 

The authors consider an important problem and attempt to study it using a large grant peer review data set from a reputable funding agency. The question of comparative reliability in ranking and rating is interesting. However, the actual quality assessment procedure that is being studied relative to standard rating is not ranking; it could be more accurately described as “tie-breaking score-based rank percentile” procedure. Therefore, the manuscript’s title and abstract claims are misleading.

Is the manuscript technically sound, and do the data support the conclusions?

There are substantial technical concerns with the manuscript that do not allow one to determine if the data support the conclusions about the use of ranking in grant peer review. The most important concerns include:

1. Although the title and abstract state that reliability of ratings and rankings are compared in the manuscript, that is not the case. Instead, the authors compare reliability of ratings and rank percentiles, obtained after reviewers were asked to break ties in their ratings. Rank percentiles are substantially different from rankings and have distinct properties. Many desirable properties of rankings, such as the direct comparison of proposals without reference to a scale, are lost when converting ranks to rank percentiles.

We have clarified our description of the process of ranking in the methodology and provided the following example in an on-line appendix. We are unsure what the reviewer means by the comment “Many desirable properties of rankings, such as the direct comparison of proposals without reference to a scale, are lost when converting ranks to rank percentiles”. The grant peer reviewers never worked with percentile ranks, only with the rank order of applications. They were asked to order all applications from best to worst, breaking ties as appropriate. The ranking process had no scale. We provide examples to illustrate how the grant peer reviewers changed the rank order of the applications and how they broke ties.

It was only after the grant peer reviewer completed the ranking exercise that CIHR converted the ranks to percentile ranks. This conversion was necessary because reviewers of the same application may have reviewed a different number of applications. The mean of the percentile ranks provided by each reviewer of the application was the final application score.

Appendix A provides three de-identified examples from the data set used in the analysis (please see attached reviewer response for clearer formatting). 

In Example 1, Reviewer A reviewed 14 applications and scored them from 18 to 106 for the highest score application. The reviewer was presented with the initial ranking based on score. Two applications were tied with a score of 70. The reviewer broke this tie providing application #6 with an adjusted rank of 8 and application #7 with a rank of 9. Reviewer A also changed the rank order of application #13 and #14, from a rank of first to application #14 to second and application #13 was ranked first. CIHR converted the adjusted ranks to rank percentiles. If reviewer was the unit of analysis the mean rank percentile, by definition, would be 50% (as illustrated). However, in our study application was the unit of analysis. To obtain an overall score the application the mean of the percentile ranks of each reviewer was calculated. In example 1, for application #6, there were 5 reviewers with initial scores ranging from 70 to 96, and percentile ranks from 41.177 to 75.00 providing an overall mean percentile rank score of 54.549. 

UNIT OF ANALYSIS IS THE REVIEWER

Example Scores and percentile ranks for REVIEWER A and the 14 applications he was assigned to:

Application number Reviewer pin Reviewers initial score Reviewer INITIAL Rank Reviewer ADJUSTED rank Reviewers percentile rank

1 A 18 13 14 0.000

2 A 48 12 13 7.692

3 A 54 11 12 15.385

4 A 60 10 11 23.077

5 A 62 9 10 30.769

6 A 70 8 8 46.154

7 A 70 8 9 38.462

8 A 74 7 7 53.846

9 A 80 6 6 61.539

10 A 84 5 5 69.231

11 A 88 4 4 76.923

12 A 92 3 3 84.615

13 A 98 2 1 100.000

14 A 106 1 2 92.308

 Mean percentile rank 50.000

UNIT OF ANALYSIS IS THE APPLICATION

Scores and percentile ranks for APPLICATION no 6 from the 5 REVIEWERS assigned to review it & Mean percentile rank:

Application number Reviewer pin Reviewers initial score Reviewers percentile rank Mean percentile rank

6 a 70 46.154 54.549

6 b 80 41.177 54.549

6 c 84 66.667 54.549

6 d 84 43.750 54.549

6 e 96 75.000 54.549

 Mean percentile rank 54.549 

In Example #2, Reviewer Z reviewed 16 applications, scoring them from 64 to 106 with one tied score for application #15 and #16 of 76. The reviewer broke this tie ordering application #15 with a rank of 11 and #16 with a rank of 12. The lowest ranked application was #11 with an adjusted rank of 16 and a percentile rank of 0.00. Application #11 had 5 reviewers who scored the application between 48 and 84 with percentile ranks ranging from 0.00 to 62.5. The mean percentile rank score for this application was 28.56.

Example Scores and percentile ranks for REVIEWER Z and the 16 applications he was assigned to:

Application number Reviewer pin Reviewers initial score Reviewer INITIAL Rank Reviewer ADJUSTED rank Reviewers percentile rank

11 Z 64 15 16 0.000

12 Z 68 14 15 6.667

13 Z 70 13 14 13.333

14 Z 72 12 13 20.000

15 Z 76 11 11 33.333

16 Z 76 11 12 26.667

17 Z 86 10 10 40.000

18 Z 88 9 9 46.667

19 Z 90 8 8 53.333

20 Z 92 7 7 60.000

21 Z 96 6 6 66.667

22 Z 98 5 5 73.333

23 Z 100 4 4 80.000

24 Z 102 3 3 86.667

25 Z 104 2 2 93.333

26 Z 106 1 1 100.000

 Mean percentile rank 50.000

Scores and percentile ranks for APPLICATION no 11 from the 5 REVIEWERS assigned to review it & Mean percentile rank:

Application number Reviewer pin Reviewers initial score Reviewers percentile rank Mean percentile rank

11 Z 64 0.000 28.560

11 P 62 13.333 28.560

11 Q 84 62.500 28.560

11 T 70 31.250 28.560

11 Y 48 35.714 28.560

 Mean percentile rank 28.559 

In Example #3, Reviewer T reviewed 13 applications, scoring them from 78 to 112. The reviewer broke two ties involving application #36 and #37 as well as #38 and #39. In addition, the reviewer changed the rank order of the top three applications, moving the third ranked to the second. The top ranked application for Reviewer T (application #43) had 5 reviewers, four of whom ranked the application as first (rank percentile=100.00) in the pool of applications they reviewed, providing an overall mean percentile rank from the 5 reviewers of 87.50 for application #43. 

Example Scores and percentile ranks for REVIEWER T and the 13 applications he was assigned to:

Application number Reviewer pin Reviewers initial score Reviewer INITIAL Rank Reviewer ADJUSTED rank Reviewers percentile rank

31 T 78 11 13 0.000

32 T 84 10 12 8.333

33 T 88 9 11 16.667

34 T 96 8 10 25.000

35 T 98 7 9 33.333

36 T 100 6 7 66.667

37 T 100 6 8 58.333

38 T 102 5 5 50.000

39 T 102 5 6 41.667

40 T 104 4 4 75.000

41 T 106 3 2 91.667

42 T 108 2 3 83.333

43 T 112 1 1 100.000

 Mean percentile rank 50.000

Scores and percentile ranks for APPLICATION no 43 from the 5 REVIEWERS assigned to review it & Mean percentile rank:

Application number Reviewer pin Reviewers initial score Reviewers percentile rank Mean percentile rank

43 T 112 100.000 97.500

43 EE 106 87.500 97.500

43 Q 100 100.000 97.500

43 A 112 100.000 97.500

43 V 104 100.000 97.500

 Mean percentile rank 97.500 

2. Relatedly, some conclusions noted by the authors are the direct mathematical result of the construction of their “ranking” data. Such conclusions include: (1) the authors note on line 240 that the mean rank percentile is 50%, which is a consequence of their data conversion that would hold for any data source. 

The mean rank percentile is, by construction, is 50% when the unit of analysis is the reviewer as shown in our three examples Our study used the application as the unit of analysis. The final score for the application was the mean of the percentile ranks provided by all reviewers of the application. As illustrated in the three examples, the mean percentile rank of the application is, of course, not 50%. 

We did not conduct any conversion of the data. The rating, ranking, adjusted ranking, and percentile ranking of each reviewer for an application, and the final mean ranking for each application is provided by CIHR. We have all ranks and percentile ranks for all reviewers for all applications for each stage of the competition. No grants were excluded from the analysis.

(2) Similarly, the lack of statistical significance for coefficients in the rank percentile model is likely the result of the “zero-sum” construction of rank percentiles, and not necessarily a true lack of statistical significance.

The zero-sum construction of rank percentiles would only apply if we were using reviewer as the unit of analysis (see above examples). We wish to point out that the zero-sum construction does not apply when using application as the unit of analysis where the outcome is the mean of all reviewers’ percentile ranks. 

3. Although authors note that “inclusion of only top scoring applications in phase 2 reduced the reliability of assessment”, they still make the conclusion on lines 356-360 that IRR was found to be smaller for the assessment of research project quality (phase 2) than that of researchers (phase 1). This comparison is flawed. That IRR decreased in phase 2 is primarily a mathematical result of removing the estimated “worst” proposals (see Erosheva, Martinkova, and Lee, 2021).

We apologize for the lack of clarity in the presentation of results. We think the reviewer has assumed that the same phase 1 score was used to calculate the IRR for applications that were successful in reaching phase 2. This is not correct. In phase 2, a different set of reviewers was selected and they used a different set of criteria and scoring system to rate the quality of the research program, using the full range of possible scores. We have revised this paragraph describing the results to simply report that the ICC was lower for phase 2. In the discussion we have outlined likely reasons; 1) that the quality of candidates selected to submit their research program to phase 2 was more homogeneous, making it more difficult for reviewers to distinguish amongst higher quality proposals, and/or 2) that there is less agreement among reviewers about what constitutes a high quality research program than there is for evaluating the quality of the candidate (that was evaluated in phase 1). 

4. On lines 138-139, the authors note “it was assumed that the distribution of the quality of applications assigned to each reviewer would be equivalent.” What is precisely meant by the phrase “the distribution of the quality of applications…would be equivalent”? Whatever the formal description of this assumption might be, it appears to be quite stringent and unlikely to hold, especially if there was substantial variability in the number of proposals assessed by each reviewer. The authors report the mean numbers of 12.6 and 9.5 of proposals assessed by each reviewer in phases 1 and 2, respectively, but do not comment on the range. Given this implausible assumption, the authors should explore its influence on the results.

In the “ideal” world, all reviewers would review each application submitted to a given program/committee, and the average rank for these applications would be assigned to the application. However this is a completely impractical. Nor is it possible to randomly allocate applications to reviewers whereby the random assignment would ensure that an equivalent number of poor, good and excellent applications would be reviewed by each reviewer. The priority in CIHR’s assignment of applications to reviewers was to optimize the match between reviewer expertise and the content of the application. A priori, neither CIHR or the chair of a committee would have knowledge of the quality of the application, therefore there was no possibility of systematic bias in assignment (for example where only the poor applications would go to one reviewer and the excellent applications to another). Therefore, it is assumed that, on balance, reviewers would obtain a mix of applications of varying quality. Any violation of this assumption would contribute to random errors in measurement and lead to an under-estimate of the reliability of ranking. We have noted this limitation in the discussion. The range in the number of applications assigned to a reviewer has been added to the description of Table 1 results.

5. Comparing the reliability (variance) of ratings and rankings is ultimately challenging given the ordinal properties of rankings. The authors’ attempts to make this comparison, while commendable in goal, are not statistically appropriate.

We include the histogram and rug plots of the log of the inter-rater variance for individual application scores and ranks for analyses reported in Table 3, to clarify that the data are normally distributed and continuous. Confusion about the nature of the data may have occurred if the reviewer assumed that the reviewer was the unit of analyses in the study. In our study, application is the unit of analysis. Each application has four to five reviewers, each of whom scored the application. 

Distribution of the Variance of Ranks between Raters of an Application (see attached reviewer response for graph)

Distribution of the Variance of Scores between Raters of an Application (see attached reviewer response for graph)

6. On numerous instances, we believe the authors use the phrase “percent rank” when referring to “rank percentile”. These are different concepts.

This was our error, and we have corrected our terminology throughout the manuscript. 

7. Interpretation of regression coefficients should be done conditionally on all other variables being held constant.

In the analysis section we indicated that multivariate regression was used to estimate the associations between application, applicant and reviewer characteristics and each outcome, and provided the regression models in the footnotes of each table. By definition, multivariate models provide estimates of the independent association of a given variable, controlling for all other variables in the model.

Is the manuscript presented in an intelligible fashion and written in standard English?

Yes.

References:

Erosheva, E. A., Martinková, P., & Lee, C. J. (2021). When zero may not be zero: A cautionary note on the use of inter‐rater reliability in evaluating grant peer review. Journal of the Royal Statistical Society: Series A (Statistics in Society)

Has the statistical analysis been performed appropriately and rigorously?

Please refer to the concerns raised in answering the previous question.

Have the authors made all data underlying the findings in their manuscript fully available?

The data will not be publicly available citing privacy concerns. We suggest the authors attempt to release a de-identified subset of the data publicly, as has been done with similar data from the Swiss NSF in Heyard et al. (2021), the National Institutes of Health in Erosheva et al. (2020), and the American Institute of Biological Sciences in Pearce and Erosheva (2022).

References:

Heyard, R., Ott, M., Salanti, G., & Egger, M. (2022). Rethinking the funding line at the Swiss national science foundation: Bayesian ranking and lottery. Statistics and Public Policy, 9(1), 110-121.

Erosheva, E. A., Grant, S., Chen, M. C., Lindner, M. D., Nakamura, R. K., & Lee, C. J. (2020). NIH peer review: Criterion scores completely account for racial disparities in overall impact scores. Science Advances, 6(23), eaaz4868.

Pearce, M., & Erosheva, E. A. (2022). A unified statistical learning model for rankings and scores with application to grant panel review. Journal of Machine Learning Research, 23(210), 1-33.

---

## [Editor Report · Decision Letter 3]

6 Sep 2023

PONE-D-22-19788R3Ranking versus rating in peer review of research grant applicationsPLOS ONE

Dear Dr. Tamblyn,

Thank you for submitting your manuscript to PLOS ONE. After careful consideration, we feel that it has merit but does not fully meet PLOS ONE’s publication criteria as it currently stands. Therefore, we invite you to submit a revised version of the manuscript that addresses the points raised during the review process.

Dear author/s, thanks for submitting your work to PLoS ONE,

All major conceptual and methodological reviews of the article have been addressed. Still, please consider the following minor revisions:

- Add a space (•) before cited references. This applies to all the manuscript (e.g., from “investigators(4-8)” to “investigators•(4-8)”

- Authors mention in section “Applicant characteristics” that “For each publication, we retrieved the citation reports and assigned the impact factor for each journal by linking the ISSN of the journal to the Journal Citation Record file. When there was no recorded ISSN, we used the full and abbreviated journal name to make the link.” It is not clear why the authors sourced the impact factor of journals if the computation of the H-Index do not require such a value (H-Index=for a set of articles N of an author and defining ci as the number of citations corresponding to an article i then ordering the set of articles in decreasing order according to the number of citations).

In addition, although at this point it might be out of the article scope as it is, would be to include an H-Index based indicator that controls for an authors number of collaborators which is the hm index (For a set of articles N with ci the number of citations for the article i and ai the number of corresponding authors, the cumulative sum of the inverse of the number of authors is proposed as the effective rank . Then, sorting the set of articles in decreasing order according to the number of citations: https://doi.org/10.1016/j.joi.2008.05.001). The advantage is to compute an H index based indicator that controls for the number of collaborators throughout a researcher's career.

- PLoS ONE strongly advocates for a data availability and reproducible results as much as possible. Authors stated in “Data availability” item: “No - some restrictions will apply” considering the Canadian Institutes for Health Research (CIHR) policy. However, it might be available by request to the Vice-President of Research Programs-Operations at CIHR. Ultimately, I strongly suggest, if possible, to the authors to share in a public/institutional repository the code scripts produced for the analysis so other authors with the approval from the Canadian Institutes for Health Research (CIHR) could be able to reproduce/replicate/crowdsource the results of the study. Sincerely, Julian D. Cortés

Associated editor

We look forward to receiving your revised manuscript.

Kind regards,

Julian D. Cortes

Academic Editor

PLOS ONE
---

## [Author Response · Author response to Decision Letter 3]

14 Sep 2023

Dear author/s, thanks for submitting your work to PLoS ONE,

All major conceptual and methodological reviews of the article have been addressed. Still, please consider the following minor revisions:

- Add a space (•) before cited references. This applies to all the manuscript (e.g., from “investigators(4-8)” to “investigators•(4-8)”

We have corrected the spacing issue throughout the manuscript.

- Authors mention in section “Applicant characteristics” that “For each publication, we retrieved the citation reports and assigned the impact factor for each journal by linking the ISSN of the journal to the Journal Citation Record file. When there was no recorded ISSN, we used the full and abbreviated journal name to make the link.” It is not clear why the authors sourced the impact factor of journals if the computation of the H-Index do not require such a value (H-Index=for a set of articles N of an author and defining ci as the number of citations corresponding to an article i then ordering the set of articles in decreasing order according to the number of citations).

This was our oversight as we had another measure we were working on for other projects that required the journal impact factor. Thank-you so much for picking up this error. We have corrected it in our revised manuscript.

In addition, although at this point it might be out of the article scope as it is, would be to include an H-Index based indicator that controls for an authors number of collaborators which is the hm index (For a set of articles N with ci the number of citations for the article i and ai the number of corresponding authors, the cumulative sum of the inverse of the number of authors is proposed as the effective rank . Then, sorting the set of articles in decreasing order according to the number of citations: https://doi.org/10.1016/j.joi.2008.05.001). The advantage is to compute an H index based indicator that controls for the number of collaborators throughout a researcher's career.

Thank you for this suggestion for a minor revision. However, based on the comment from a previous peer review, we have decided to keep the number of collaborators as a separate variable that we adjust for in the models to transparently address the reviewer’s concerns about endogeneity. 

Prior Reviewer: Another concern is the statistical treatment. In particular, the authors have missed potential problems associated with endogeneity in the data. Endogeneity broadly refers to situations in which an explanatory variable (e.g., the H-Index of the applicant) correlates with the error term of the regression equation (tables 4 and 5). In this case, if authors do not control for how many co-authors an applicant has worked with in previous works, this might increase the error term of the regression equation because H-index correlates with the number of co-authors (Vinkler, 2023).

Our response to the prior review: This is a very good point. We have redone the analysis to include the number of unique collaborators listed in the applicant’s publications to address this problem of unmeasured covariates in the error term. The correlation between the H index and number of collaborators is reasonably high (r=0.5). Both the H index and number of collaborators are significantly associated with rater variance and the score in the same direction. However, there is no impact of including number of collaborators on estimates of applicant, reviewer and review characteristics on variance and bias. The new results are presented in Table 4 and 5.

- PLoS ONE strongly advocates for a data availability and reproducible results as much as possible. Authors stated in “Data availability” item: “No - some restrictions will apply” considering the Canadian Institutes for Health Research (CIHR) policy. However, it might be available by request to the Vice-President of Research Programs-Operations at CIHR. Ultimately, I strongly suggest, if possible, to the authors to share in a public/institutional repository the code scripts produced for the analysis so other authors with the approval from the Canadian Institutes for Health Research (CIHR) could be able to reproduce/replicate/crowdsource the results of the study.

We have noted that the SAS code used to conduct the analyses will be available at McGill -Dataverse curated by the McGill University Library (McGill University Dataverse (borealisdata.ca) within the next 2 weeks

Sincerely,

Julian D. Cortés

Associated editor

---

## [Editor Report · Decision Letter 4]

18 Sep 2023

Ranking versus rating in peer review of research grant applications

PONE-D-22-19788R4

Dear Dr. Tamblyn,

We’re pleased to inform you that your manuscript has been judged scientifically suitable for publication and will be formally accepted for publication once it meets all outstanding technical requirements.

Kind regards,

Julian D. Cortes

Academic Editor

PLOS ONE

Additional Editor Comments (optional):

I am satisfied with the revisions carried out based on earlier feedback. Therefore, I recommend your paper for acceptance, subject to the usual final formatting checks required by the editorial office.
---

## [Editor Report · Acceptance letter]

25 Sep 2023

PONE-D-22-19788R4 

Ranking versus rating in peer review of research grant applications 

Dear Dr. Tamblyn:

I'm pleased to inform you that your manuscript has been deemed suitable for publication in PLOS ONE. Congratulations! Your manuscript is now with our production department. 

Kind regards, 

on behalf of

Professor Julian D. Cortes 

Academic Editor

PLOS ONE